

# Trends and spatial variation in rain-on-snow events over the Arctic Ocean during the early melt season

Tingfeng Dou[1], Cunde Xiao[2], Jiping Liu[3], Qiang Wang[4], Shifeng Pan[5], Jie Su[6], Xiaojun Yuan[7], Minghu Ding[8], Feng Zhang[5], Kai Xue[1], Peter. A. Bieniek[9], Hajo Eicken[9]

[1]College of Resources and Environment, University of Chinese Academy of Sciences, Beijing 100049, China.

[2]State Key Laboratory of Earth Surface Processes and Resource Ecology, Beijing Normal University, Beijing 100875, China.

[3]Department of Atmospheric and Environmental Sciences, University at Albany, State University of New York, Albany, NY, USA.

[4]Alfred Wegener Institute Helmholtz Centre for Polar and Marine Research, Bremerhaven, Germany.

[5]College of atmospheric science, Nanjing University of Information Science & Technology, Nanjing 210044, China.

[6]Physical Oceanography Laboratory, Ocean University of China, 238 Songling Road, Qingdao 266100, China.

[7]Lamont-Doherty Earth Observatory, Columbia University, 61 Route 9W, Palisades, NY 10964, USA.

[8]Institute of Tibetan Plateau and Polar Meteorology, Chinese Academy of Meteorological Sciences, Beijing 100081, China.

[9]International Arctic Research Center, University of Alaska Fairbanks, Fairbanks, AK 99775-7340, USA.

*Correspondence to*: T. Dou (doutf@ucas.ac.cn)



**Abstract.** Rain-on-snow (ROS) events can accelerate the surface ablation of sea ice, thus greatly influencing the ice-albedo

feedback. However, the variability of ROS events over the Arctic Ocean is poorly understood due to limited historical station

data in this region. In this study early melt season ROS events were investigated based on four widely-used reanalysis

products (ERA-Interim, JRA-55, MERRA2 and ERA5) in conjunction with available observations at Arctic coastal stations.

The performance of the reanalysis products in representing the timing of ROS events and the phase change of precipitation

was assessed. Our results show that ERA-Interim better represents the onset date of ROS events in spring and ERA5 better

represents the phase change of precipitation associated with ROS events. All reanalyses indicate that ROS event timing has

shifted to earlier dates in recent decades (with maximum trends up to -4 to -6 days/decade in some regions in ERA-Interim),

and that sea ice melt onset in the Pacific sector and most of the Eurasian marginal seas is correlated with this shift. There has

been a clear transition from solid to liquid precipitation, leading to more ROS events in spring, although large discrepancies

were found between different reanalysis products. In ERA5, the shift from solid to liquid precipitation phase during the early

melt season has directly contributed to a reduction in spring snow depth on sea ice by more than -0.5cm/decade averaged

over the Arctic Ocean since 1980, with the largest contribution (about -2.0cm/decade) in the Kara-Barents Seas and

Canadian Arctic Archipelago.

## 1 Introduction

Changes in the phase of precipitation (solid or liquid) can impact the freeze-thaw processes of cryospheric components (such

as sea ice, snow and permafrost), the hydrological cycle, and terrestrial and marine ecosystems. With the rapid warming of

the Arctic climate, precipitation will increasingly occur in liquid form (Bintanja et al., 2017). Liquid precipitation helps the

growth and northward expansion of vegetation and promotes the ablation of snow, ice and permafrost (e.g. Putkonen and

Roe 2003; Rennert et al. 2008; Casson et al. 2010). Increased frequency of liquid precipitation in spring can accelerate the

thawing of permafrost, which in turn leads to more methane release (Neumann et al., 2019). The snowmelt associated with

rain-on-snow (ROS) events can directly lead to a decrease in spring snow water equivalent and have a significant influence

on water storage and supply in snowmelt-controlled areas (Birsan et al., 2005; Renard et al., 2008; Jeong et al. 2016).

Weather stations are relatively sparse in the Arctic region, and very few of them have sensors that can distinguish between

solid and liquid precipitation (Peterson et al., 2006; White et al., 2007; Rawlins et al., 2010). A recent study based on station

observations indicated that spring precipitation over the Arctic landmass has transitioned from solid precipitation to liquid

precipitation in recent decades (Han et al. 2018). Because there are no long-term observations over Arctic sea ice, few

studies have examined precipitation phases over the sea ice region so far. Screen and Simmonds (2012) analyzed the

seasonal variations of snowfall and rainfall over the Arctic Ocean and showed that the fraction of summer precipitation

falling as snow has decreased in recent decades. Dou et al. (2019) analyzed changes in the phase of precipitation over coastal

sea ice in northern Alaska. They found that since the 1990s ROS events have been shifting to earlier dates in May, helping

trigger and accelerate surface ablation of sea ice in the region. In contrast, solid precipitation (snowfall) in spring can retard

sea ice melt to some extent (Perovich et al. 2017).

Atmospheric reanalysis data are often used to understand climate change processes and to drive sea ice-ocean models, which

warrants an assessment of how well ROS events are represented in reanalyses. A few previous studies (Screen and

Simmonds, 2012; Lindsay et al., 2014; Boisvert et al., 2018) have examined Arctic precipitation characteristics in reanalysis

products, but there is a lack of systematic studies of the variations of precipitation phases over the Arctic Ocean. In particular, analysis during early stages of sea ice ablation (March to June) is lacking. In this period ROS events play a key role in initiating snow and sea ice melt, because the occurrence, timing and quantity of rainfall can greatly affect reductions in snow

albedo, enhance heat transfer into the snow pack, and promote the formation and development of melt ponds.

This study is motivated by the need to improve understanding of changes in the phase of precipitation during the early stages of sea ice ablation (March to June) and to evaluate the timing of ROS events promoting onset of melt during this period. Due to the extremely limited coverage of historical in-situ observations over the Arctic Ocean we first assess the representation of

ROS events in four state-of-the-art reanalysis datasets using the limited long-term observations available. The station observations were derived from a single station along the Alaska coast and 14 stations in the Canadian Arctic Archipelago (CAA) located in close vicinity to sea ice. Second, we use the reanalysis datasets to investigate changes in ROS events over the Arctic Ocean in recent decades. Both the timing and amount of liquid precipitation in the early melt season were analyzed. We also consider the question as to whether such shifts in early melt season precipitation are part of an Arctic-

wide trend or a more localized phenomenon, and to what extent the ROS events influence variations in snow depth and enhance sea ice surface ablation.

## 2 Materials and Methods

### 2.1 Precipitation in ERA-I, MERRA2, JRA-55 and ERA5

Gridded precipitation information over the Arctic Ocean was derived from four reanalysis products, the European Centre for

Medium-Range Weather Forecasts (ECMWF) reanalysis interim (ERA-I, Dee et al. 2011), Modern-Era Retrospective analysis for Research and Applications (MERRA version 2, Rienecker et al., 2011), Japanese 55-year Reanalysis (JRA-55, Kobayashi et al., 2015) and the latest reanalysis product of ECMWF, ERA5 (Hersbach and Dee, 2016). No direct observations of precipitation have been assimilated into these reanalysis products. The representation of precipitation, however, can be influenced indirectly by other assimilated fields. For example, satellite measurements of microwave

radiances were used to adjust humidity fields in ERA-I, which can thereby influence precipitation indirectly (personal

communication with D. P. Dee, S. M. Uppala and A. J. Simmons). To facilitate a comparison with station observations, the

reanalysis data were bilinearly interpolated to a common 0.125º × 0.125º grid and the grid points nearest to each station were

chosen. Daily mean precipitation data were used in our analysis. We chose 0.5 mm per day as the threshold to determine the

occurrence of rainfall. This value is close to the field measurement accuracy (Dou et al., 2019) and high enough to eliminate

spurious counts of rainfall events.

## 2.2 Satellite-derived sea ice concentration

There must be snow cover on the ground or ice surface when ROS events occur (McCabe et al., 2007). Surface-based

observations and satellite remote sensing studies by Warren et al. (1999), Webster et al. (2014) and Kwok et al. (2020)

showed that most of the Arctic sea ice surface is covered by snow during March through June, with snow depth decreasing

rapidly throughout June and bare ice appearing along the marginal seas of the Arctic Ocean in July. In recent years, removal

of snow through melt has shifted into June (Kwok et al., 2020). Therefore, we used the monthly sea ice concentration (SIC)

in May and June to mask the range of ROS events over the Arctic Ocean. The SIC data from the U.S. National Snow and Ice

Data Center (NSIDC) derived from the NASA Team algorithm (Fetterer et al., 2017) is used to define the boundaries of

Arctic sea ice during the study period. SIC in this dataset is derived from passive microwave brightness temperatures. Since

the SIC data are available every 2 days before 1987, the series since 1988 is used for the analysis in this study. The original

data are on a polar stereographic grid with a spatial resolution of 25 km× 25 km; we re-gridded them onto a 0.125º × 0.125º

grid to be consistent with the format of the precipitation reanalysis data. The sea ice extent is calculated from SIC using a

threshold of 15%.

## 2.3 Station precipitation data

To evaluate the reanalysis data, the long-term records from the few available coastal stations were used. We employed total

precipitation and snowfall observations from January 1952 to June 2017 at the Utqiaġvik Weather Service Office (WSO)

Airport station, located at Utqiaġvik near the coast of the Chukchi Sea (available from the Alaska Climate Research Center,

http://climate.gi.alaska.edu/acis_data). The snowfall data are given as snow water equivalent. The snowfall amount was subtracted from the total precipitation to obtain the rainfall amount.


We also obtained total precipitation and rainfall station data across the Canadian Arctic Archipelago from the daily network program "DLY04" (part of Environment Canada's national archive) through personal communication with Julian Morales at Environment and Climate Change Canada. The DLY04 data have been quality-controlled following the current standards of Environment Canada. We selected 14 meteorological stations with relatively long timespans (1979-2007) next to sea ice in

northern Canada (north of 60°N, details shown in Table 1). The criterion used to determine the occurrence of rainfall for station data is the same as that used for reanalysis data.

**2.4 Satellite-derived sea ice melt onset dates**

At present, melt onset detection from passive microwave satellite data is based on the temporal variability of brightness temperatures at 19GHz and 37GHz. Snow and ice emissivity increases significantly with increasing wetness, i.e., as the

liquid water builds up in the snowpack and at the ice surface due to onset of melt (Markus et al., 2009). The sea ice melt onset data set was retrieved from the satellite microwave radiometer data for the Scanning Multichannel Microwave Radiometer, Special Sensor Microwave/Imager, and Special Sensor Microwave Imager and Sounder (Markus et al., 2009; Stroeve et al., 2014). This dataset has been shown to represent the melt signal of ice and snow and has been used to reveal the mechanisms triggering Arctic sea ice ablation (e.g., Mortin et al., 2016). Melt onset is described by two different

variables: early melt onset (EMO) and continuous melt onset (Markus et al., 2009). We applied the EMO criterion—the first time melt is detected–in this study, because it has been shown that this parameter is closely linked to the atmospheric processes triggering melt (Mortin et al., 2016). For the correlation analysis the native 25×25km EMO data grid for the period 1980-2017 was interpolated to the 0.125º × 0.125º reanalysis grids described in the previous section.

## 2.5 Methods

The ratio of rain/total precipitation (RPR) was analyzed to reflect the change in precipitation phase. The larger the ratio, the

greater the proportion of rainfall in the total precipitation. The increase in RPR indicates the trend from snowfall to rainfall.

The linear trends of RPR and the timing of the first rainfall event in spring were computed using the least-squares method

(Belington and Robinson, 2003) and the corresponding confidence levels (that is, the probabilities of linear trends with a

non-zero slope) were estimated by Student's t-statistic (Box et al., 2005).


The relative contribution of the change in precipitation phase to the increase in precipitation occurring as ROS events was

calculated using the method developed by Bintanja (2018), specifically his expression (10) in the supplementary information.

Snow depth on sea ice is very sensitive to changes in precipitation phase. Solid precipitation increases snow depth while

liquid precipitation does not. We estimated the contribution of precipitation phase transitions to the trend (interdecadal

variation) in snow depth on sea ice over the period 1980-2017. We first detrended the variations in precipitation phase (RPR),

and then multiplied the total precipitation by the detrended RPR to obtain the detrended precipitation that fell as ROS events

over the past decades. Then, the decrease in the snowfall amount due to the change in precipitation phase was derived from

the differences between the original and the detrended precipitation occurring as ROS events. The snowfall reduction (snow

water equivalent) was converted to snow depth reduction based on the climatological monthly mean snow density given by

Warren et al. (1999). Finally, the linear trend of the variations in snow depth caused by precipitation phase change was

calculated.

## 3 Results

### 3.1 Trends in the timing of melt season ROS events

The first ROS events that occur during the spring melt season were evaluated to assess their linkage with the onset of sea ice

melt. Before analyzing the variability in timing of these events, the four reanalysis datasets were first evaluated using

precipitation measurements at the Utqiaġvik station in northern Alaska and 14 coastal stations adjacent to sea ice in the CAA (station details shown in Table 1). On average, the dates (i.e. day of the year) of the first spring ROS events in ERA-I (149±15) are slightly earlier than the station observed mean value (152±17), but closer than the other reanalysis products that

have a larger negative bias (JRA-55 (133±21); ERA5 (135±21); MERRA2 (138±17)). The standard deviation (± value) represents the spatial variability of the timing of the first spring ROS amongst the 15 stations.

Trends of the first ROS event date in northern Alaska and the CAA region show significant spatial differences across the station observations (Fig. 1a). Most stations have negative trends indicating earlier ROS events with time and there is a large

spread in the trend magnitude among the station observations. In general, the directions of the trends in ERA-I are most consistent with the station observations, and the magnitudes of the trends are comparable to the observations for two thirds of the stations (Utqiaġvik, Chesterfield Inlet, Arviat, Kugaaruk, Whale Cove, Repulse Bay, Taloyoak, Paulatuk, Komakuk Beach). ERA5 is consistent with ERA-I in the trend direction except at Igloolik, but it underestimates the trend magnitudes at some stations (Fig. 1a). MERRA2 is in line with most stations in the direction of the trends except at three stations

(Kugaaruk, Rankin Inlet, Taloyoak). For trend values, MERRA2 is comparable to observations at nearly half of the stations (Fig. 1a). JRA-55 exhibits a relatively large deviation in both the directions and magnitudes of the trends from these station observations.

ERA-I, JRA-55 and MERRA2 have similar spatial patterns over the Arctic Ocean for the trend of the date of the first ROS

events (Fig. 2). These products reveal trends towards earlier ROS events across most of the study areas except for the North Atlantic-Arctic region, Bering Sea and Hudson Bay (Fig. 2a-c). In contrast, ERA5 presents a trend towards earlier ROS events in nearly all regions including the Atlantic sector (Fig. 2d). All reanalysis datasets show consistent trends over the Arctic Ocean but there are discrepancies in the location of significant trends among the datasets. The negative trend in ERA-I reaches -4 to -6 days/decade in some regions of the Beaufort, East Siberian and Laptev Seas and stays below -2

days/decades in most other parts of the Arctic Ocean. JRA-55 has a stronger trend than ERA-I and other products, especially in the Eurasian Basin where the trend at most grid points can be up to -6 to -8 days/decade. The magnitude of the trend in

MERRA2 is smaller than that in JRA-55 and ERA-I. The strongest trends over the Arctic Ocean in MERRA2 range from -2 to -4 days/decade and are mainly located over the marginal seas. ERA5 has a smaller area with significant trends although it presents the most significant trend (more than -8 days/decade) in the east-central part of the Canada basin. In summary,

although there are some discrepancies, all reanalysis products consistently show that the first ROS events during the spring season have been occurring earlier over many regions of the Arctic Ocean and the most pronounced trends were over the marginal seas.

## 3.2 Sensitivity of sea ice melt onset to ROS events

Earlier studies explained potential trigger mechanisms for sea ice ablation mainly in terms of atmospheric physical processes.

Specifically, melt-triggering weather patterns were shown to be associated widely with intensified atmospheric transient eddy activity and enhanced northward transport of warm and moist air (Hegyi and Deng, 2017). As a result, there are typically positive anomalies in air temperature, precipitable water vapor and cloud fraction which increase the downward longwave radiative flux at the surface, contributing to initial melt (Mortin et al., 2016; Oltmanns et al., 2019; Huang et al., 2019). In addition, ROS events may also occur alongside warm and moist air invasions (Gimeno et al., 2016) and influence

the snow and ice ablation. ROS events can directly lead to an increase in the amount of liquid water at the surface and alter the emissivity. On the other hand, rainfall can effectively reduce the surface albedo and bring in additional heat when it penetrates into the snow layer, initiating positive snow/ice albedo feedback (Dou et al., 2019).

We examined the ERA-I data set, which was found to compare well with the observations at the Alaska Arctic and CAA

stations in the preceding section, to analyze the sensitivity of early sea ice melt onset to ROS events in the Arctic. A detrended correlation analysis reveals that EMO is sensitive to ROS events in the Pacific sector of the Arctic Ocean and most of the Siberian marginal seas (Fig. 3). EMO in the Kara, Laptev and Chukchi Seas, along with the eastern part of the East Siberian Sea exhibits the highest correlation with and therefore is most sensitive to ROS events (Fig. 3). There are also significant correlations over the northern waters of the Chukchi Sea, the western central Arctic Ocean, Hudson Bay, and

waters north of Severnaya Zemlya, but the FRD occurs after EMO in these areas. In conclusion, the sensitivity of EMO to

FRD is mainly present in the marginal seas of the Arctic Ocean.

The above analysis provides further evidence to explain increased vulnerability of Arctic sea ice to the climate change. Rapid

sea ice loss is not just driven by warm weather systems which cause positive anomalies of heat flux, moisture convergence

and downward long-wave radiation flux (e.g., Kug et al., 2010; Lee et al., 2017), thus leading to rapid melting of sea ice (e.g.,

Parkinson et al., 2013; Serreze et al., 2016; Praetorius et al., 2018; Bi et al., 2019). We suggest that springtime ROS events

are also a factor influencing sea ice melt onset although their impact is more pronounced on regional scales, as shown in Fig.

3.

### 3.3 Variability and trends in precipitation phase

ROS events can accelerate snow and ice melt through the release of latent heat, determined by the amount of rainfall (Dou et

al., 2019). The amount of rainfall depends on the total precipitation and the portion of total precipitation occurring as rainfall,

as quantified by the rain precipitation ratio, RPR (see section 2.5). Many prior studies based on historical data and model

projections have shown increases in total precipitation over the Arctic region to varying degrees toward the end of the

twenty-first century (e.g., Walsh et al., 1998; Kattsov et al., 2000; Min et al., 2008; Collins et al., 2013; Bintanja et al., 2014).

In our study we evaluated and compared changes in RPR across four different reanalysis products.

The RPR averaged over the Arctic Ocean is overall higher for all spring months in ERA-I than in the other three reanalysis

datasets (Fig. 4). MERRA has the lowest RPR among the four reanalysis datasets (Fig. 4c). JRA-55 and ERA5 have similar

RPRs in March, April, and June, while May RPR based on ERA5 is lower than that in JRA-55 (Fig. 4b and d). RPR is

relatively small in March and April in all reanalyses, indicating that most precipitation falls as snow during this period. RPR

increases significantly in May and June, and rainfall accounts for about half of the total precipitation by June, but there are

large discrepancies between datasets. The interannual variability of RPR in JRA-55 is significantly larger than those in the

other three products in June.

RPR in the four products exhibits consistently increasing trends in each month over past decades, especially in May and June.

The most significant trend occurs in June for all the reanalysis datasets (Fig. 4). Averaged over the whole Arctic sea ice area,

the RPR trend in June in ERA-I, JRA, MERRA and ERA5 amounts to 2.6, 2.8, 2.4 and 2.1%/decade, respectively. The RPR

trends in May from ERA-I, JRA, MERRA2 and ERA5 are 1.7, 1.1, 0.8, and 1.3 %/decade, respectively. We note that the

changes in the RPR in June/May over the studied period in all the products, although statistically significant, are smaller than

the spread of the RPR mean values among these reanalysis products.

We further assessed the reanalysis datasets using the station observations in northern Alaska and CAA.  Due to the lack of

significant trends in RPR in March and April (Fig. 4), we focus on the RPR in May and June in the following, which is the

period when precipitation phase matters for snow/ice ablation. Averaged over the stations, the mean RPR in June in ERA5

(65.7%±4.0%) and JRA-55 (66.1%±11.5%) is closer to the observations (63.3%±9.1%), but it is overestimated by ERA-I

(88.3%±7.9%) and underestimated by MERRA2 (47.9%±6.9%). In May, the observed mean RPR at the stations

(13.1%±2.3%) is well reproduced by ERA5 (15.4%±2.6%), while JRA-55 (30.7%±2.1%) overestimates it and MERRA2

(8.9%±1.8%) underestimates it. Actually, the ERA-I overestimates the RPR in all months from March to June (not shown),

which is consistent with earlier studies (Leeuw et al., 2015; Wang et al., 2019).


For RPR in June, ERA-I and ERA5 have similar trend directions, and reproduce the observed direction at all stations except

for Repulse Bay (Fig. 1b). JRA-55 also captures the direction of the trends except at three stations (Kugaaruk, Igloolik, and

Repulse Bay). ERA5 has trend magnitudes comparable with the observations at 13 of the stations (except for Arviat and

Repulse Bay), and ERA-I is comparable with 11 of the station observations (Fig. 1b). JRA-55 is comparable to the

observations at more than half of the stations, while MERRA2 performs relatively poorly, with trend values comparable to

the observations only at six stations. For the trend of RPR in May, the four products are able to reproduce the trend direction

at most stations (Fig. 1c). Similar to the situation in June, ERA5 performs better than the other three products in northern

Alaska and CAA in May.

The spatial patterns of the trends in June RPR obtained from the four reanalysis datasets are shown in Figure 5a-d. It can be

seen that increasing trends of RPR exist over most of the Arctic sea ice area, although there are significant spatial variations.

ERA-I has the largest area with significant increasing trends, including the sector of the East Siberian Sea that extends to the

center of the Arctic Ocean (ESAO) and the Kara Sea, where the increasing trends range up to 4-6%/decade (Fig. 5a). It also

has a broader and significant increasing trend close to sea ice edges in the Atlantic sector. In JRA-55, high increasing trends

also exist over the ESAO region, but the area with statistically significant values is smaller than in ERA-I (Fig. 5b). Another

region with high increasing trends in JRA-55 is the Canadian Basin (Fig. 5b). The significant increasing trends in MERRA2

are mainly located in the marginal seas including the eastern Canada Basin, Chukchi, Laptev and Kara Seas (Fig. 5c). ERA5

has significant increasing trends in the Beaufort Sea, north Chukchi Sea, and the Kara Sea, and there is a weakly negative

trend over the eastern Canada Basin, which is different from the other three datasets (Fig. 5d). Although the spatial patterns

are quite different, the maximum trend values (4-6%/decade) are similar in the four reanalysis datasets.

In May, significant RPR trends (4-6 %/decade) are present in the western Arctic Ocean in both ERA-I and JRA-55, while

they are also present in the western Kara Sea in ERA-I (Fig. 5e-f). The spatial pattern of the RPR trend in May in ERA5 is

close to MERRA2, but the magnitude of the trends is greater than in MERRA2 over the marginal seas (Fig. 5g-h). Overall,

the spatial patterns of RPR trends differ considerably among the products in both months, but they have significant

increasing trends with similar maximum values in certain regions.

**3.4 Contribution of changes in precipitation phase to snow reduction over sea ice**

The trend of rainfall during ROS events accumulated from March to June was estimated based on ERA5, which shows the

best performance in the representation of RPR when compared with station observations. The trends of total precipitation

and RPR were also analyzed for the same period. As shown in Figure 6, the ROS shows a significant increasing trend toward

higher rainfall amounts in the marginal seas and CAA area except for the southern part of the Chukchi Sea and the western

part of the East Siberian Sea. In the same period, the total precipitation amounts significantly increased in the northern

Barents Sea and parts of the Kara Sea. They significantly declined along the eastern Greenland, and the change in other areas

of the Arctic Ocean is relatively small. The trend of RPR is generally consistent with the trend of ROS (Fig. 6). Further

analysis indicated that the increase in precipitation occurring as ROS events was mainly caused by changes in precipitation

phases (increased RPR): On average, more than 80% of the increase in early melt season precipitation occurring as rainfall

can be attributed to the solid-to-liquid precipitation phase transition, and the remainder by the increase in total precipitation.

Using ERA5 we estimated the contribution of changes in precipitation phases to trends in spring snow depth on sea ice over

the period 1980-2017. Here we only consider the direct contribution from the mass of precipitation (less snowfall, lower

snow thickness) while the indirect contribution from the latent heat of rainfall is beyond the scope of this study. Our analysis

indicates that the impact on snow depth by precipitation phase changes has significant spatial variations (Fig. 7). The phase

change leads to declines in snow depth in most of the Arctic marginal seas. In some small areas (including the central

Canada Basin and part of the East Siberian Sea) increases in snow depth are induced. The Kara/Barents Seas and Canadian

Arctic Archipelago got contributions of the largest decreasing trend (more than -2.0cm/decade). Averaged over the Arctic

Ocean, the reduction rate in snow thickness associated with precipitation phase transition is -0.5cm/decade over the past

decades. The actual contribution of the changes in the precipitation phases should be greater if the latent heat of increased

rainfall is taken into account. This study suggests that the interdecadal decrease in snow depth on sea ice in spring is

enhanced by the change in precipitation phase (solid to liquid) during the initial ablation period, in addition to the impact

from delayed sea ice freeze-up during autumn (Webster et al., 2014).

## 4 Discussion and conclusions

Observations on landfast ice in the Chukchi Sea showed that spring rain-on-snow (ROS) events have an important impact on

the sea ice ablation process during the early melt season (Dou et al., 2019). In particular, the timing of the first ROS events

of the melt season is a key factor influencing the surface melt onset. However, because continuous precipitation observations

are not available in the Arctic Ocean, there is little knowledge about the timing of the first ROS events in the Arctic sea ice

area. This study, for the first time, synthesizes station observations at coastal sites in Arctic North America and multiple

atmospheric reanalysis datasets. We assessed the timing of the first ROS events in spring and the fraction of total precipitation occurring as rain over the Arctic Ocean during the initial phase of the melt season in four reanalysis products.

Our results show that the date of the first ROS events in ERA-I is closer to the station observations than in the other three products, in terms of the average timing, interannual variability and trends. ERA-I and JRA-55 have similar trend magnitudes for the timing of the initial spring ROS events over the Arctic Ocean, with maximum values up to -4 to -6 days/decade. The trend values and the area with significant trends in MERRA2 and ERA5 are smaller than in the other two datasets. The multiple reanalysis products consistently indicate that trends towards earlier spring ROS events exist

throughout much of the Arctic Ocean over the past decades, with the most pronounced negative trends in the marginal seas. Results further demonstrate that sea ice melt onset is sensitive to the timing of the first melt season ROS events in the Pacific sector of the Arctic Ocean and the Eurasian marginal seas, especially over the Chukchi Plateau, in the Kara, Laptev and East Siberian Seas.

The rain-precipitation-ratio (RPR) averaged over the Arctic Ocean shows an increasing trend in all months through March-June in all reanalysis datasets, although there are differences in the magnitude of the trend among the datasets. The RPR value and its increasing trend averaged over the Arctic Ocean are strongest in May and June in all datasets, and RPR in ERA-I is significantly higher than in other datasets over all spring months. For the mean value of RPR in June and May, ERA5 is closer to the observations at coastal stations, followed by JRA-55, while ERA-I overestimates and MERRA

underestimates the observations. ERA-I and ERA5 are basically the same in the representation of trend directions, which are closest to the observations. ERA5 performs better than ERA-I in representing the RPR trend strength. JRA-55 does not perform as well as ERA, but it also captures the trend direction and strength at more than half of the stations.

The June RPR trends are positive over most of the Arctic Ocean in all reanalysis datasets, and their maximum trend values

(4-6%/decade) are similar although the spatial patterns are different. The trend strength in May is much weaker than in June, and the area with significant trends is reduced, especially in MERRA2. In June, both ERA-I and JRA-55 show significant
increasing trends in the ESAO region, while MERRA shows significantly positive trends in the marginal seas, with area smaller than in ERA-I and JRA-55. The significant increasing trend in ERA5 is found in the Beaufort Sea, the northern Chukchi Sea and Kara Sea in June. In May, there are significant trends (4-6 %/decade) in the western Arctic Ocean in both ERA-I and JRA-55, while the trends in ERA5 and MERRA2 are weaker.


ERA5 more reasonably reproduces the observed RPR and its trends than ERA-I compared with station observations. Several new techniques have been incorporated into ERA5 since ERA-I that have likely improved its performance. Firstly, ERA5 applies a prognostic cloud microphysics scheme, with separate cloud liquid, cloud ice, rain and snow prognostic variables (Sotiropoulou et al., 2015), which is more realistic than the scheme used in ERA-I that determines liquid and ice in cloud only by a temperature threshold (e.g. Dutra et al., 2011). Secondly, ERA5 uses much higher spatial and temporal resolutions, to improve the ability of the model to simulate meteorological conditions on regional scales, which is especially beneficial for simulating precipitation. In addition, ERA5 uses a newer assimilation scheme and involves various newly reprocessed datasets, for example, the reprocessed version of the Ocean and Sea Ice Satellite Application Facilities sea ice concentration (OSI-SAFr), and recent instruments that could not be ingested into ERA-I. As a result, ERA5 has a more consistent sea surface temperature and sea ice concentration (Hersbach and Dee, 2016), which will also improve the precipitation simulation over areas with strong and frequent air-sea interaction.



This study suggests that the solid-to-liquid precipitation phase transition (i.e. increased ROS events) contributed to a substantial reduction in snow depth on sea ice during the early melt season. RSO events have other important impacts that are not studied in this paper. For example, the formation of superimposed ice as a result of ROS events, can also accelerate sea ice surface ablation during the early melt season by promoting the formation of melt ponds (Eicken et al., 2004; Petrich et al., 2012) and strengthening the ice-albedo feedback with potential to cause greater ice mass loss in the warm period (Perovich et al., 1997; Stroeve et al., 2014; Schröder et al., 2014).



As atmospheric reanalysis datasets are often used to drive ocean-ice models and for understanding climate dynamics, it is crucial to understand the uncertainties in the timing, phase, and spatial distribution of precipitation in these datasets. Among the studied datasets, ERA-I most realistically represents the timing of ROS events, and ERA5 favorably reproduces the RPR during ROS events and the phase change of precipitation in the study period. All the reanalysis datasets have certain biases

compared to individual station observations. Besides requiring new techniques in different reanalysis systems to better reproduce precipitation, reliable observations that can better confine reanalysis are also required in the future.

**Data availability**

The precipitation data at the Utqiaġvik Weather Service Office (WSO) Airport station can be accessed through the Alaska Climate Research Center, http://climate.gi.alaska.edu/acis_data. (last access: 28 July, 2020).

**Author contribution**

T. Dou, C. Xiao, H. Eicken and J. Liu jointly conceived the study. T. Dou analyzed the data and wrote the manuscript with additional input from C. Xiao, H. Eicken, Q. Wang, J. Su, X. Yuan, and P. A. Bieniek. All of the authors discussed the results and contributed to interpretations.

**Competing interests:**

The authors declare no competing interests.

**Acknowledgments**

We wish to acknowledge ECMWF for ERA-Interim and ERA5 data. JRA-55 reanalysis data are obtained from the Japan Meteorological Agency Climate Prediction Division, Global Environment and Marine Department. MERRA-2 reanalysis data are available from the NASA Goddard Earth Sciences Data and Information Services Center (GES-DISC). This study is

funded by the National Key Research and Development Program of China (2018YFC1406103), and the National Nature Science Foundation of China (NSFC 41971084).



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





**Table 1.** Stations in Canadian Arctic Archipelago (CAA) selected for comparison with reanalysis datasets

| WMO ID | Climate ID | Station Name | | Lat (N) | Lon (W) | Elevation (m) |
|---|---|---|---|---|---|---|
| | 2300707 | Chesterfield Inlet | Nunavut | 63.347 | 90.731 | 10 |
| | 2300MKF | Arviat | Nunavut | 61.1 | 94.067 | 10 |
| | 2303092 | Kugaaruk | Nunavut | 68.541 | 89.797 | 16 |
| 71083 | 2303401 | Rankin Inlet | Nunavut | 62.817 | 92.117 | 32 |
| | 2303986 | Whale Cove | Nunavut | 62.24 | 92.598 | 12 |
| 71917 | 2401200 | Eureka | Nunavut | 79.983 | 85.933 | 10 |
| | 2402540 | Igloolik | Nunavut | 69.383 | 81.8 | 21 |
| 71909 | 2402594 | Iqaluit | Nunavut | 63.75 | 68.55 | 22 |
| | 2403490 | Repulse Bay | Nunavut | 66.521 | 86.225 | 23 |
| 71580 | 2403854 | Taloyoak | Nunavut | 69.55 | 93.583 | 27 |
| | 2203057 | Paulatuk | Northwest Territories | 69.361 | 124.075 | 5 |
| | 2203912 | Tuktoyaktuk | Northwest Territories | 69.433 | 133.026 | 4 |
| | 2502501 | Ulukhaktok | Northwest Territories | 70.763 | 117.806 | 36 |
| 71969 | 2100685 | Komakuk Beach | Yukon Territory | 69.583 | 140.183 | 7 |



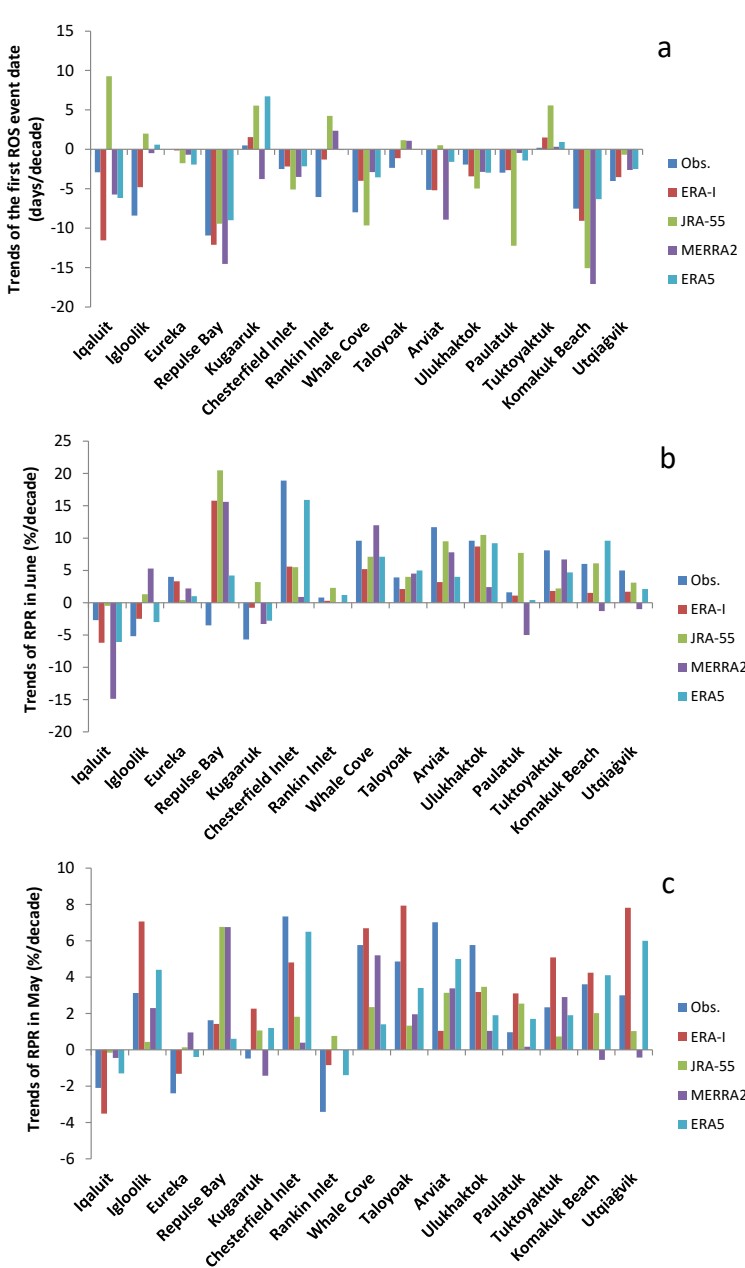

**Figure 1: (a) Comparison of the trends in the first rain-on-snow (ROS) event date between the four reanalysis datasets and the station observations. (b) Comparison of trends in the rain-precipitation-ratio (RPR) in June between different reanalysis datasets and the station observations. (c) Same as (b) but for May. The Utqiaġvik station and 14 stations in the Canadian Arctic Archipelago (CAA, see Table 1 for station information) are used and arranged from east to west.**


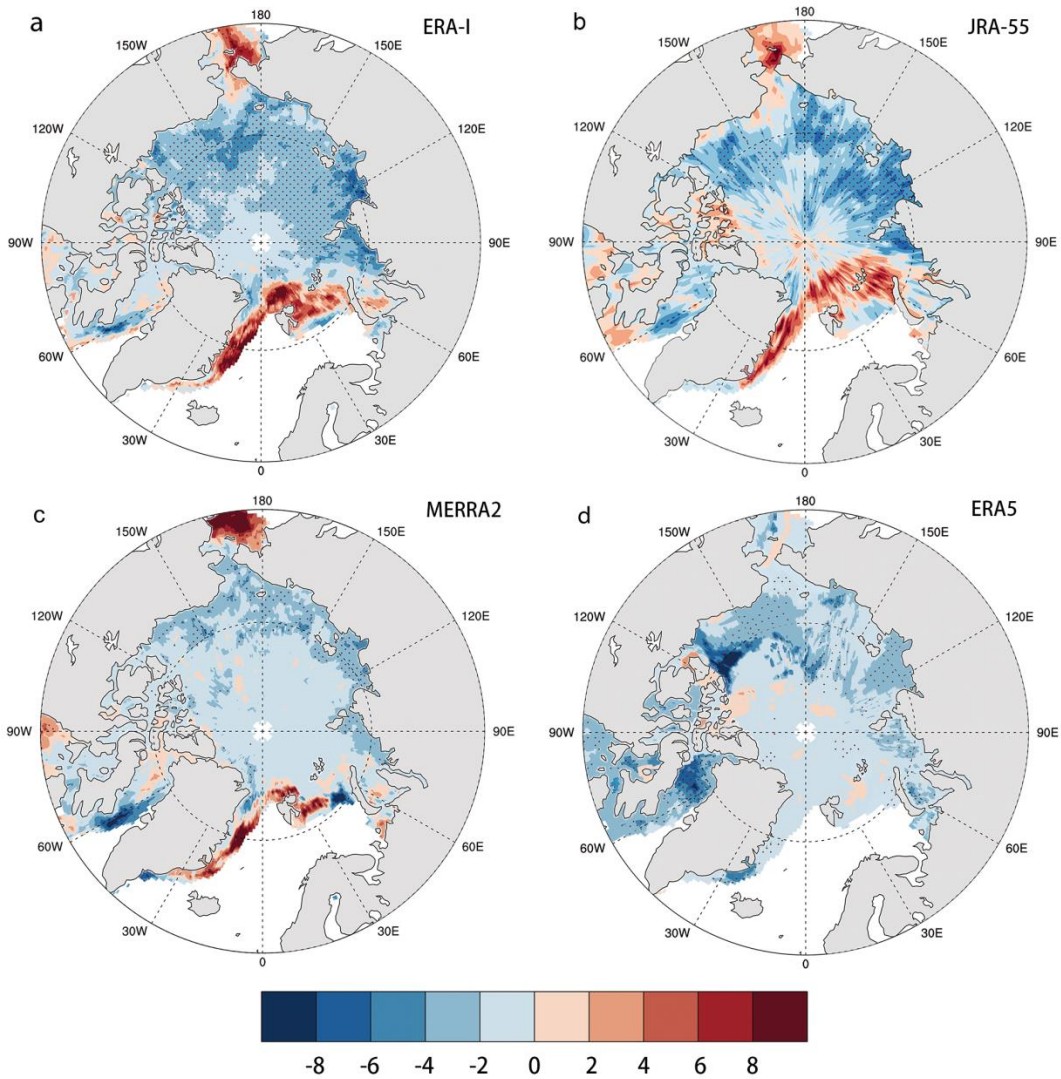

**Figure 2: Trend in the date of the first rainfall in spring (March to June) over Arctic sea ice north of 50°N during 1980-2017 in (a) ERA-I, (b) JRA-55, (c) MERRA2 and (d) ERA5 (Units: days/decade). The trend is calculated only for grid cells which experienced rainfall between March and June in more than 80% of the years in the record (i.e., in >30 years). Dotted regions indicate that the trends are significant at the 95% confidence level or higher (p< 0.05). A negative trend means that the first rainfall shifts to earlier dates.**



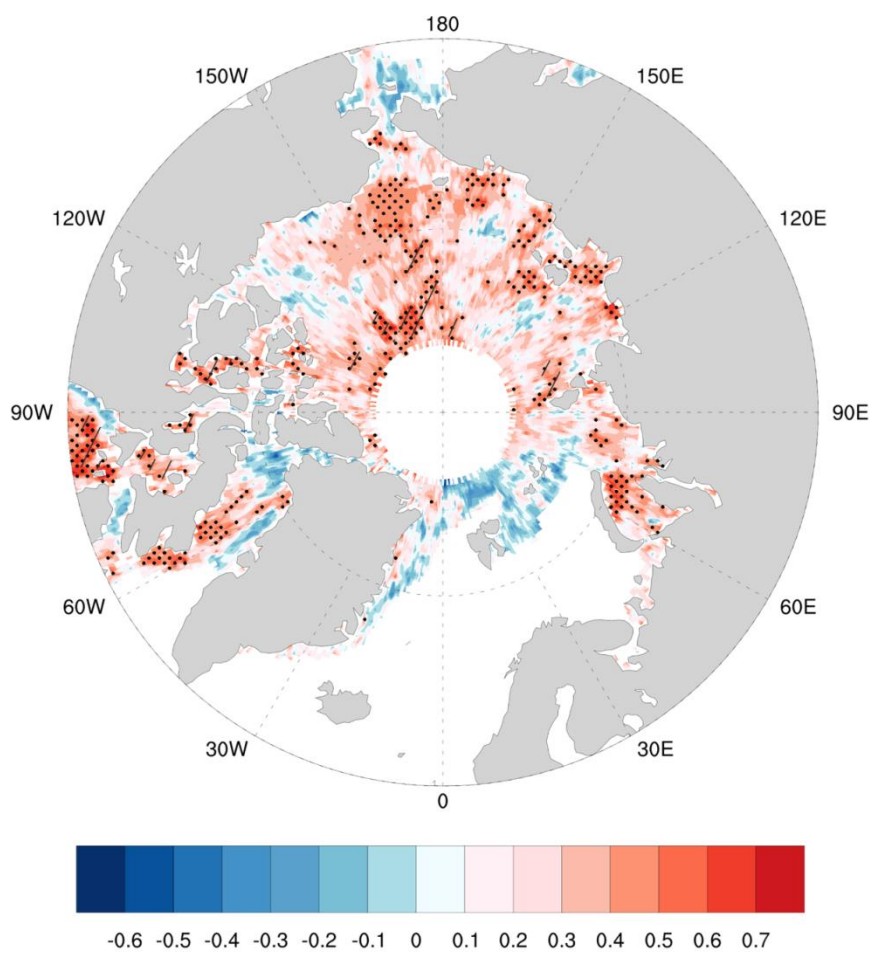


**Figure 3: The spatial distribution of correlations between linearly detrended first ROS event date (FRD) and early melt onset date (EMO) over 1980–2017. Regions where the correlation coefficients pass the 95% confidence level (p< 0.05) are denoted by black dots. Regions where the FRD comes after EMO are also denoted by slash.**





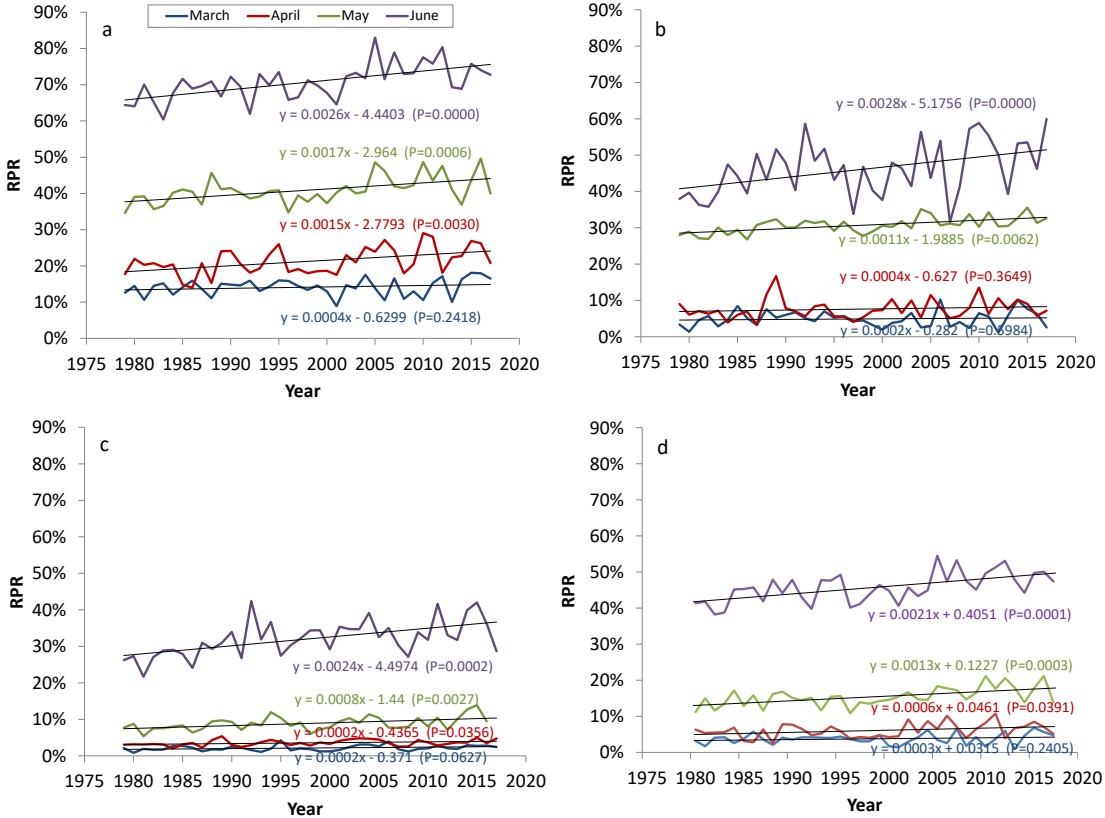

Figure 4: Time series of monthly rainfall precipitation ratio (RPR) averaged over the Arctic Ocean for (a) ERA-I, (b) JRA-55, (c) MERRA and (d) ERA5.





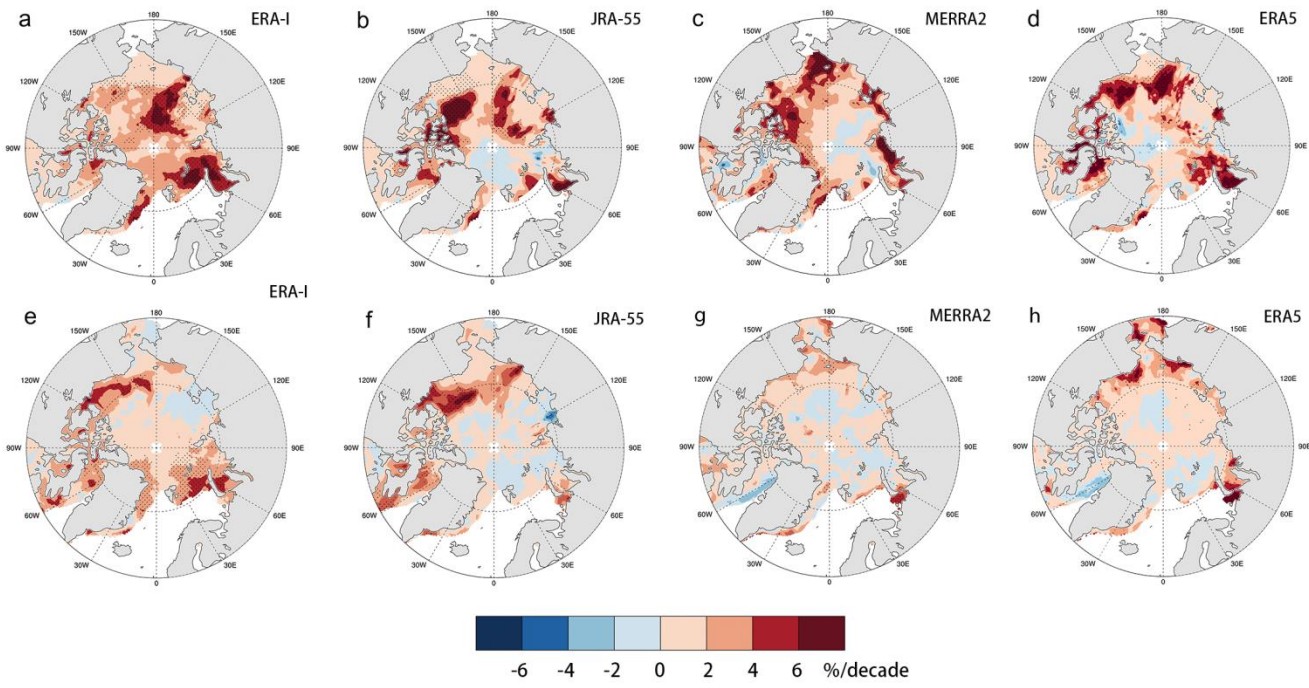

**Figure 5: Linear trend of rainfall precipitation ratio (RPR) in June (upper panel) and May (bottom panel) over Arctic sea ice during 1980-2017 in ERA-I (a,e), JRA-55 (b,f), MERRA2 (c,g) and ERA5 (d,h). Regions passing the 0.05 significance test are denoted by black dots.**





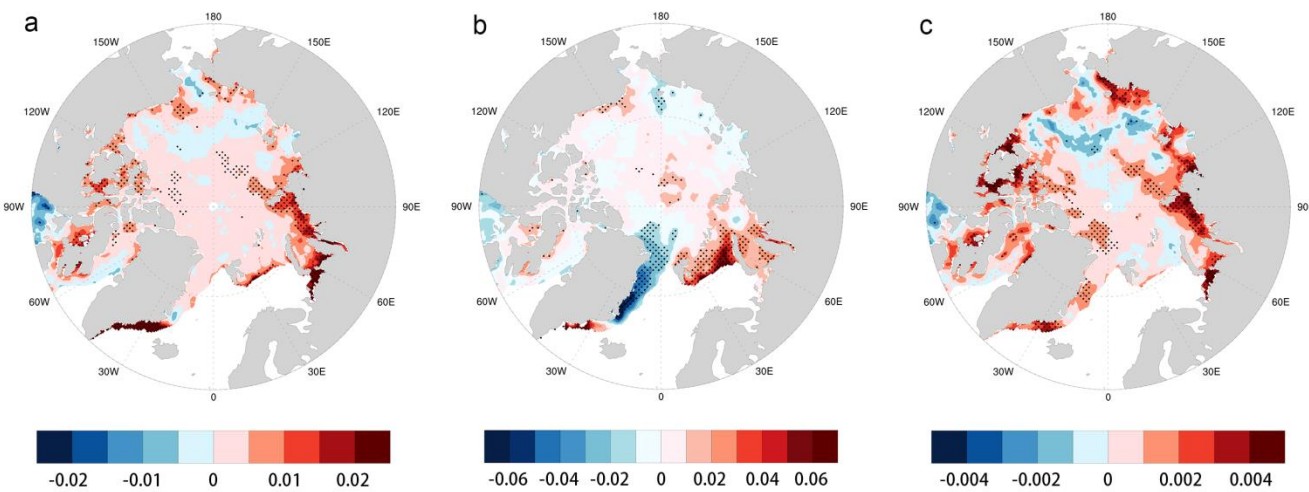

**Figure 6: Trend of rainfall (a), total precipitation (b) and RPR (c) during March and June based on ERA5 over Arctic sea ice**

515 **during 1980-2017. Regions passing a 0.05 significance test are denoted by dots. The trend is derived from the slope of a linear**

**regression. Unit of a and b: cm/decade; unit of c: %/decade.**



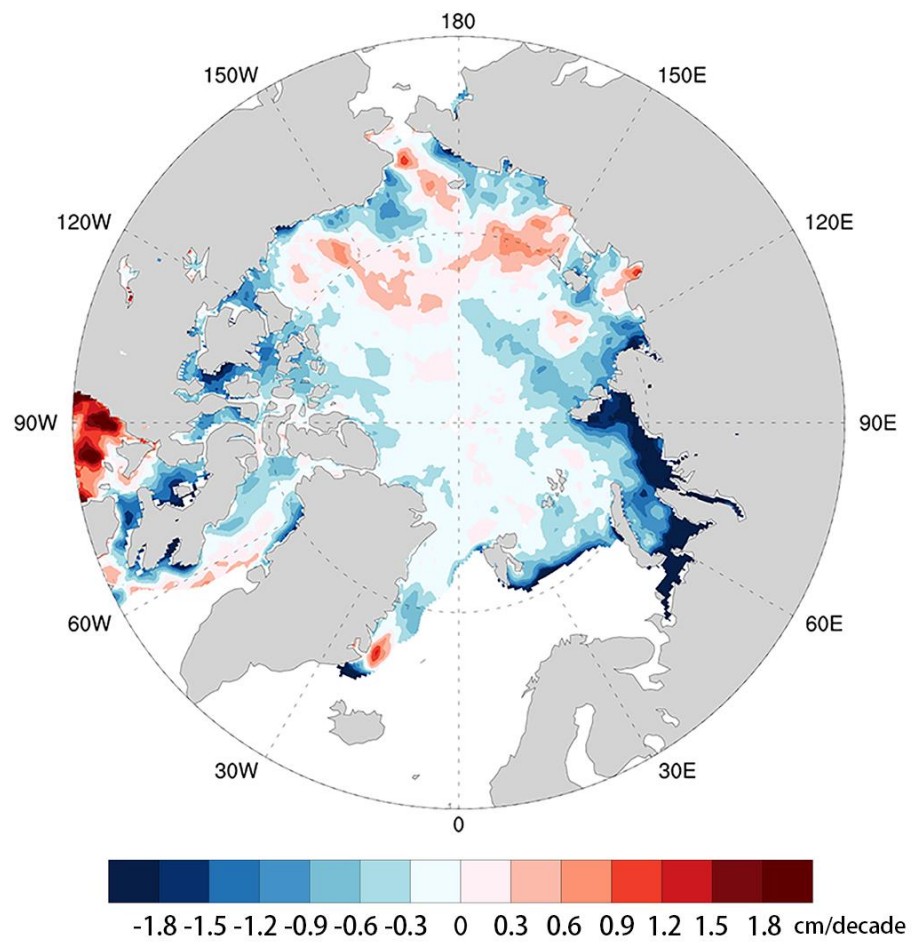

**Figure 7: The interdecadal trends in snow depth on sea ice induced by the precipitation phase transition in the early melt season**

520    **during 1980-2017**