# Peer review of "Trends and spatial variation in rain-on-snow events over the Arctic Ocean during the early melt season"

_The Cryosphere, 2020_

## Referee Comment (RC1) · Anonymous Referee #1 · 25 Sep 2020

This study uses several reanalyses to study the trends and spatial variation of rainfall on snow-covered sea ice in March-June over the Arctic Ocean and how these events relate to early melt onset. They also compare reanalysis results to observations at several coastal weather stations to assess the validity of reanalysis output. The key points, as I interpret them, are: • Rain-on-snow events are often a trigger for early melt onset on sea ice. • Rain events on snow-covered sea ice are occurring earlier in the season. This is partly because of a shift from solid to liquid precip phase • The shift in rainfall-to-precipitation ratio is a measurable cause of declining snow depth on sea ice (-0.5 cm/decade averaged over Arctic Ocean). • The ERA products (ERA-Interim and ERA5) are generally better matches with the observations, and ERA5 is

the most consistent for rainfall-precipitation ratios. I have several comments about the structure of the text and points of clarification, but most of the writing is already clear. The figures are clear and illustrate the points well. There is one aspect of the methods that I think needs to be better described, but assuming I've interpreted it correctly, the methods seem sensible and appropriate. I think this will be an excellent fit for The Cryosphere journal after some revisions to improve the clarity, the discussion, and the textual organization.

Section 1 Comments Line 38: I believe the Rennert piece cited here was actually published in 2009 – the May 1 issue of Journal of Climate.

I think this section has a good cross-section of papers on ROS, precipitation phase, and sea ice. However, I think it is worth including some more literature on sea ice melt onset. Much of this literature appears in the results section already. There are other good papers to cite for the impact of warm, moist air transport leading to melt onset for sea ice, such as Kapsch et al. (2016) and Liu & Schweiger (2017).

Section 2 Comments Line 77-78: There is assimilation of land-surface precipitation in MERRA-2, although is tapers off to 0 weighting at 62.5°N (Reichle et al., 2017). Therefore, it would be more accurate to specify that for the location of interest in the Arctic Ocean there is no direct assimilation of precipitation data. Line 83-85: I am not convinced that 0.5 mm/day is sufficient to remove all spurious precipitation events for MERRA-2. Excessive precipitation is a worse problem in MERRA-2 than ERA-Interim or JRA-55 (Boisvert et al., 2018; Figure 10). For ROS detection at an hourly scale, Crawford et al. (2020) used 0.1 in./event (2.54 mm/event) as a threshold with MERRA-2 (Figure 2). I know Bieniek et al. (2018) used 0.254 mm/day, but they were also using ERA-Interim, not MERRA-2. I'm not sure what the impact changing the threshold would have on results, but the fact that MERRA-2 has relatively poor matches with observations makes me think a higher threshold would help make them align better. That is what Crawford et al. (2020) concluded, and it might work here. Given that a) the authors have other reanalyses to work with and b) I don't think other additional

analysis is necessary, my advice is to acknowledge this possibility in the text rather than re-do the analysis with a higher threshold. Line 106-111: I'm not entirely sure how the station data from Environment Canada would be acquired by another researcher to reproduce that results of this study. It is stated these data are part of the national archive, but also that they were acquired via "personal communication". Is there no link or DOI or citable information for the national archive data? Line 131 – 132: The method described by Bintanja (2018) involves a discrete difference between two periods. It's not stated what two periods are being used for that differencing in this study. Line 133 – 142: This description also merits some clarification. For example, the meaning of the word "original" in Line 139 is unclear to me. Do the authors mean "original" with respect to time (i.e., "original" means the early part of the time series)? Do the authors mean the time series of precipitation occurring as ROS prior to detrending? I would think they mean the latter because of previous statements about detrending. Another thing that is unclear is that "decrease in snowfall" sounds like an overall rate (mm of snow/yr), whereas saying "differences" (pluralized) implies subtracting two time series element-wise. The latter appears more in line with the rest of the description.

Section 3 Comments Lines 148-150: I'm not convinced that difference is a problem. ROS events can lead to percolation and re-freezing in the snowpack during the cold season, so it seems likely the first ROS detected sometimes pre-dates melt onset, leading to lower average values for first ROS event day than for early melt onset day. Line 184: Since the Gimeno reference doesn't seem to focus on ROS events, I wonder if Bieniek et al. (2018) might be a better reference here. It may be focused on terrestrial Alaska, but it explicitly links ROS events to moisture transport like atmospheric rivers. Given the author list here, I assume there is some familiarity. Line 186: I think that statement about emissivity alteration merits a reference, however logical. I know it's described in Markus and Cavalieri (2000), but there may be a better reference. Also Line 186: Rather than "On the other hand", I think "Additionally" is a more appropriate phrase here. Organization of Section 3.2 and Line 205: I think Line 205 (the latent heat argument) would be more effective in the explanation for why ROS matters to melt

onset, which is explored in the prior section. Additionally, Section 3.2 is short enough, that I think the authors would be better off by basically flipping the order of the first two paragraphs. Paragraphs 1 and 3 cover similar ground and could be condensed a little if consecutive. Another reason for this is that the last line of Section 3.3 makes sense more immediately following the current paragraph 1 (179-187) than following paragraph 2 (189-196). Line 251: "Canadian Basin" is used here, but "Canada Basin" is used elsewhere. Lines 260-261: I think the authors can use stronger language here – in all eight cases, the trends are predominantly positive, and that's not clear from the current statement. Lines 269-272: This statement goes by fast, and the authors don't fully explain what "further analysis" was done. Looking back at the methods section, this must be the Bintanja (2018) method. A brief reminder here would go a long way. Line 280: I am not sure what the phrasing "got contributions of the largest decreasing trend" means. Do the authors simply mean these regions "exhibit" the largest decreasing trend"? Line 285: A follow-up of sorts to the Webster et al. (2014) paper also explored the importance of cyclone activity in driving snow depth (Webster et al., 2019). This might be a good place to discuss that. Here or perhaps in the introduction.

Section 4 Comments Lines 291-292: In isolation, this statement is not true because other studies have combined coastal station observations and renalyses in North America to detect rain-on-snow events (e.g., Rennert et al., 2009; Bieniek et al., 2018; Crawford et al., 2020). I believe the spirit of this statement is that this is the first study to synthesize these datasets to examine rain on snow-covered sea ice. I agree that the application to sea ice is novel, but tweaking the language slightly would be best. Lines 295 – 320: These paragraphs are all strictly summary of the results and add little or no new discussion. They are therefore less effective than the first paragraph or last three paragraphs of this section. I encourage the authors to condense them into one paragraph that highlights one or two key points in a more generalized way (i.e., shifting closer to the language of the abstract). Lines 305-306: In Lines 227-228, the authors mention a "lack of significant trends" for March and April, so it seems incongruous to mention them here as increasing trends. I would restate this

as "May and June" instead of "March-June". It also might make sense to remove "The RPR value and its increasing trend averaged over the Arctic Ocean are strongest in May and June in all datasets," if only May and June are discussed. Lines 322-346: I think these paragraphs are stronger than the earlier part of this section. The first sentence of the penultimate paragraph in particular is a clear, useful summative sentence. But the length of this discussion/conclusions section buries that statement. Works Cited in this Review Bieniek, P. A., Bhatt, U. S., Walsh, J. E., Lader, R., Griffith, B., Roach, J. K., & Thoman, R. L. (2018). Assessment of Alaska rain-on-snow events using dynamical downscaling. Journal of Applied Meteorology and Climatology, 57(8), 1847–1863. http://doi.org/10.1175/JAMC-D-17-0276.1 Boisvert, L. N., Webster, M. A., Petty, A. A., Markus, T., Bromwich, D. H., & Cullather, R. I. (2018). Intercomparison of Precipitation Estimates over the Arctic Ocean and Its Peripheral Seas from Reanalyses. Journal of Climate, 31(20), 8441–8462. http://doi.org/10.1175/JCLI-D-18-0125.1 Crawford, A. D., Alley, K. E., Cooke, A. M., & Serreze, M. C. (2020). Synoptic Climatology of Rain-on-Snow Events in Alaska. Monthly Weather Review, 148, 1275–1295. http://doi.org/10.1175/MWR-D-19-0311.s1 Kapsch, M.-L., Graversen, R. G., Tjernström, M., & Bintanja, R. (2016). The effect of downwelling longwave and shortwave radiation on Arctic summer sea ice. Journal of Climate, 29(3), 1143–1159. http://doi.org/10.1175/JCLI-D-15-0238.1 Liu, Z., & Schweiger, A. (2017). Synoptic conditions, clouds, and sea ice melt-onset in the Beaufort and Chukchi Seasonal Ice Zone. Journal of Climate, 30, 6999–7016. http://doi.org/10.1175/JCLI-D-16-0887.1 Markus, T., & Cavalieri, D. J. (2000). An enhancement of the NASA Team sea ice algorithm. IEEE Transactions on Geoscience and Remote Sensing, 38(3), 1387–1398. http://doi.org/10.1109/36.843033 Reichle, R. H. Q. Liu, R. D. Koster, C. S. Draper, S. P. P. Mahanama, and G. S. Partyka, (2017). Land surface precipitation in MERRA- 2. Journal of Climate, 30, 1643–1664, https://doi.org/10.1175/JCLI-D- 16-0570.1. Rennert, K. J., Roe, G., Putkonen, J., & Bitz, C. M. (2009). Soil thermal and ecological impacts of rain on snow events in the circumpolar Arctic. Journal of Climate, 22(9), 2302–2315. http://doi.org/10.1175/2008JCLI2117.1 Webster, M. A., Parker, C., Boisvert, L.,

& Kwok, R. (2019). The role of cyclone activity in snow accumulation on Arctic sea ice. Nature Communications, 10(1), 1–12. http://doi.org/10.1038/s41467-019-13299-8

---

## Referee Comment (RC2) · Anonymous Referee #2 · 11 Oct 2020

Summary

This study evaluates the performance of four reanalysis datasets (ERA-I, JRA-55, MERRA2, and ERA5) in representing the timing of ROS events and the phase change of precipitation during the spring melt season over the Arctic Ocean. Comparing with observations at 15 Arctic coastal weather stations, the authors find that the date of the first ROS events in ERA-I is closer to the observations than that in the other three products, while ERA5 better represents the phase change of precipitation associated with ROS events in spring. The study then investigates trends and spatial variations of ROS events and rain-precipitation-ratio (RPR) over the Arctic Ocean during the melt

season. The results show trends towards earlier spring ROS events over most of the Arctic Ocean in recent decades, with the most negative trends in the marginal seas. There has been a clear transition from solid to liquid precipitation over the Arctic Ocean from Match to June, consistent with more ROS events in spring.

General Comments

This is an interesting study and fits well to The Cryosphere. Overall the paper reads well, however, I have one major concern, which is the weak justification of the use of ERA-I and ERA5 for trend analyses. In addition, the structure of the paper is not always clear. Outlined in the comments below are some suggestions that will hopefully improve the final version of the paper.

Why the four reanalysis products (ERA-I, JRA-55, MERRA2, and ERA5) are chosen for the study? Especially MERRA2 instead of MERRA? Boisvert et al. (2018) suggested that precipitation in MERRA was realistic but there were large biases in MERRA2. In their abstract: "When compared with drifting ice mass balance buoys, three reanalyses (ERA-Interim, MERRA, and NCEP R2) produce realistic magnitudes and temporal agreement with observed precipitation events, while two products [MERRA, version 2 (MERRA-2), and CFSR] show large, implausible magnitudes in precipitation events."

Observations from 15 coastal station are used to assess the performance of the reanalysis in this study. In my opinion, the observations at the coastal stations may be representative of nearby marginal seas, but unlikely to be representative of the central Arctic Ocean. Therefore the justification of using ERA-I for ROS events and ERA5 for RPR analyses is rather weak in the paper. The authors should at least provide a summary of evaluation results from previous studies, i.e. Boisvert et al. (2018). In addition, it is critical for the reanalysis products to be consistent over time (1980-2017) for trend analyses. Please verify this in the paper.

Specific comments

Section 2.1, Please provide a bit more detail about the reanalysis products, such as the resolutions. Map in Fig. 2(b) is somewhat blurry, is that due to the coarser resolution of JRA-55 relative to the other three products?

L83-84, are the results sensitive to the 0.5 mm threshold used to determine the occurrence of rainfall? I suggest the authors do some tests on this if haven't already. Ideally, the results shouldn't be too sensitive to the threshold.

L97-98, "The sea ice extent is calculated from SIC using a threshold of 15%", please provide a reference.

Section 2.3, Please explain why observations at other weather stations are not included, such as those along the coast of European Arctic.

Fig.2, it would be helpful to provide some comments about the positive/later trends occurring in (a)-(c), but not in (d).

Section 3.2, I'd suggest to move the first paragraph to the Introduction. I think this would make the paper tighter and make it easier for readers to have a better understanding of the linkages between different components of the paper.

Fig.3, I wonder if the significant correlations between EMO and ROS are due to the fact that they are both correlated with surface air temperature (SAT)? SAT was used to indicate melt onset on sea ice in previous studies. Dou et al. (2019) suggested that the year-to-year variability of the timing of first spring ROS was closely tied to the timing of persistent warming events.

L195, FRD is not defined.

Section 3.3, I'd suggest move the first paragraph to the Introduction, see above.

L317, ESAO is not defined.

L321-330, "ERA5 more reasonably reproduces the observed RPR and its trends than ERA-I compared with station observations. Several new techniques have been incorporated into ERA5. . .", this seems to be in contrast with the large negative bias for first EOS events in ERA5 relative to ERA-I and observations shown earlier (L150), can you explain why?

Fig.5, I'd prefer to have maps in May on the upper panel and June on the bottom panel.

Please include the name of the reanalysis product used for the results in Fig.3 and 7 in the captions.
* * *

---

## Author Comment (AC1) · 9 Dec 2020

Response letter Dear Editor, We have studied the valuable comments from yourself and the reviewers carefully, and made further revisions in the manuscript that address your and the reviewers' concerns. Our detailed response to the reviewers' comments follows below. - Response to Reviewer #1's comments: This study uses several reanalyses to study the trends and spatial variation of rainfall on snow-covered sea ice in March-June over the Arctic Ocean and how these events relate to early melt onset. They also compare reanalysis results to observations at several coastal weather stations to assess the validity of reanalysis output. The key points, as I interpret them, are:

[Figure]

Rain-on-snow events are often a trigger for early melt onset on sea ice. Rain events on snow-covered sea ice are occurring earlier in the season. This is partly because of a shift from solid to liquid precip phase. The shift in rainfall-to-precipitation ratio is a measurable cause of declining snow depth on sea ice (-0.5 cm/decade averaged over Arctic Ocean). The ERA products (ERAInterim and ERA5) are generally better matches with the observations, and ERA5 is the most consistent for rainfall-precipitation ratios. I have several comments about the structure of the text and points of clarification, but most of the writing is already clear. The figures are clear and illustrate the points well. There is one aspect of the methods that I think needs to be better described, but assuming I've interpreted it correctly, the methods seem sensible and appropriate. I think this will be an excellent fit for The Cryosphere journal after some revisions to improve the clarity, the discussion, and the textual organization. Section 1 Comments Line 38: I believe the Rennert piece cited here was actually published in 2009 – the May 1 issue of Journal of Climate. I think this section has a good cross-section of papers on ROS, precipitation phase, and sea ice. However, I think it is worth including some more literature on sea ice melt onset. Much of this literature appears in the results section already. There are other good papers to cite for the impact of warm, moist air transport leading to melt onset for sea ice, such as Kapsch et al. (2016) and Liu & Schweiger (2017). Response: Thank you for your reminder. Yes, the Rennert piece cited here was published in 2009, and we have corrected this reference in the revised MS. According to your suggestion, several additional references are added in the discussion of the impact of warm, moist air transport on sea ice melt onset. Please see details at L225 in the revised MS with traces of changes. Kapsch M. L, Graversen RG, Tjernström M, Bintanja R (2016). The effect of downwelling longwave and shortwave radiation on Arctic summer sea ice. Journal of Climate, 29,1143–1159. https://doi.org/10.1175/JCLI-D-15-0238.1. Liu, Z. , and Schweiger, A. (2017). Synoptic conditions, clouds, and sea ice melt onset in the beaufort and chukchi seasonal ice zone. Journal of Climate, 30(17), 6999-7016. Persson, P. O. G. (2012), Onset and end of the summer melt season over sea ice: Thermal structure and surface energy perspective from SHEBA, Clim. Dyn., 39(6), 1349– 1371,

doi:10.1007/s00382-011-1196-9.

Section 2 Comments Line 77-78: There is assimilation of land-surface precipitation in MERRA-2, although is tapers off to 0 weighting at 62.5 N (Reichle et al., 2017). Therefore, it would be more accurate to specify that for the location of interest in the Arctic Ocean there is no direct assimilation of precipitation data. Response: Thank you for your suggestion. We have revised the statement here (L102). "There is no direct assimilation of precipitation data in the Arctic Ocean (Dee et al., 2011; Reichle et al., 2017)."

Line 83-85: I am not convinced that 0.5 mm/day is sufficient to remove all spurious precipitation events for MERRA-2. Excessive precipitation is a worse problem in MERRA-2 than ERA-Interim or JRA-55 (Boisvert et al., 2018; Figure 10). For ROS detection at an hourly scale, Crawford et al. (2020) used 0.1 in./event (2.54 mm/event) as a threshold with MERRA-2 (Figure 2). I know Bieniek et al. (2018) used 0.254 mm/day, but they were also using ERA-Interim, not MERRA-2. I'm not sure what the impact changing the threshold would have on results, but the fact that MERRA-2 has relatively poor matches with observations makes me think a higher threshold would help make them align better. That is what Crawford et al. (2020) concluded, and it might work here. Given that a) the authors have other reanalyses to work with and b) I don't think other additional analysis is necessary, my advice is to acknowledge this possibility in the text rather than re-do the analysis with a higher threshold. Response: Thank you for your comment. We rechecked the data used in this work, and found that we were actually looking at MERRA and not MERRA-2 in this study. We mistakenly wrote it as MERRA-2 in the original MS. As show in Figure R1, MERRA-2 gives a lower RPR because it overestimates the snowfall too much and underestimated the rainfall in the Arctic Ocean. We corrected this in the revised MS and added a discussion about the results of Boisvert et al. (2018). Please see details at L84-91 in the revised MS with trace of changes. Figure R1ïïjŽTime series of monthly rainfall precipitation ratio (RPR) averaged over the Arctic Ocean for MERRA and MERRA-2.

As you have mentioned, changing the threshold of precipitation events may affect the determination of the date of the first rainfall to a certain extent, especially for the reanalysis with more frequent trace precipitation, such as MERRA and MERRA-2 (Boisvert et al. al., 2018). Therefore, a lower precipitation threshold may result in an earlier ROS event in these reanalysis datasets, while a higher threshold may result in a later ROS event. However, this will not change the pattern of the trend of ROS occurrence time. We have acknowledged this possibility and added a discussion in the revised MS (L181-186). "Changing the threshold of precipitation events may affect the determination of the date of the first rainfall to a certain extent, especially for the reanalysis with more frequent trace precipitation, such as MERRA and MERRA-2 (Boisvert et al. al., 2018). Therefore, a lower precipitation threshold may result in an earlier ROS event in these reanalysis datasets, while a higher threshold may result in a later ROS event. However, different thresholds will not have a fundamental impact on the spatial distribution of the trend of the first rainfall timing."

Line 106-111: I'm not entirely sure how the station data from Environment Canada would be acquired by another researcher to reproduce that results of this study. It is stated these data are part of the national archive, but also that they were acquired via "personal communication". Is there no link or DOI or citable information for the national archive data? Response: Thank you for your suggestion. We have included a link for the station precipitation data across the Canadian Arctic Archipelago (http://climate.weather.gc.ca/index_e.html) in the revised MS.

Line 131 – 132: The method described by Bintanja (2018) involves a discrete difference between two periods. It's not stated what two periods are being used for that differencing in this study. Response: The statement: ".......more than 80% of the increase in early melt season precipitation occurring as rainfall can be attributed to the solid-to-liquid precipitation phase transition...." has been removed in the revised MS since it will not affect the main conclusion of this study.

Line 133–142: This description also merits some clarification. For example, the meaning of the word "original" in Line 139 is unclear to me. Do the authors mean "original" with respect to time (i.e., "original" means the early part of the time series)? Do the authors mean the time series of precipitation occurring as ROS prior to detrending? I would think they mean the latter because of previous statements about detrending. Another thing that is unclear is that "decrease in snowfall" sounds like an overall rate (mm of snow/yr), whereas saying "differences" (pluralized) implies subtracting two time series element-wise. The latter appears more in line with the rest of the description. Response: Thank you for your suggestions. The "original" here means the time series prior to detrending. We have clarified and revised this in the revised MS. ". . . the differences in the snowfall amount due to the change in precipitation phase was derived by the difference of the time series of precipitation occurring as ROS events before and after detrending."

Section 3 Comments Lines 148-150: I'm not convinced that difference is a problem. ROS events can lead to percolation and re-freezing in the snowpack during the cold season, so it seems likely the first ROS detected sometimes pre-dates melt onset, leading to lower average values for first ROS event day than for early melt onset day. Line 184: Since the Gimeno reference doesn't seem to focus on ROS events, I wonder if Bieniek et al. (2018) might be a better reference here. It may be focused on terrestrial Alaska, but it explicitly links ROS events to moisture transport like atmospheric rivers. Given the author list here, I assume there is some familiarity. Response: Thank you for your comment. The reference here has been changed to Bieniek et al. (2018) in the revised MS.

Line 186: I think that statement about emissivity alteration merits a reference, however logical. I know it's described in Markus and Cavalieri (2000), but there may be a better reference. Also Line 186: Rather than "On the other hand", I think "Additionally" is a more appropriate phrase here. Response: Thank you for your suggestion. We have added two references to support this statement. Ferraro, R. R., Peters-Lidard, C. D., Hernandez, C., Turk, F. J., Aires, F., Prigent, C., Lin, X., Boukabara, S.-A., Furuzawa, F. A., Gopalan, K., Harrison, K. W., Karbou, F., Li, L., Liu, C., Masunaga, H., Moy, L., Ringerud, S., Skofronick-Jackson, G. M., Tian, Y., Wang, N.-Y.: An Evaluation of Microwave Land Surface Emissivities Over the Continental United States to Benefit GPM-Era Precipitation Algorithms, IEEE Transactions on Geoscience and Remote Sensing, 2013(51), 378-398, 2013. Markus, T., and Cavalieri, D. J.: An enhancement of the NASA Team sea ice algorithm, IEEE Transactions on Geoscience and Remote Sensing, 38(3), 1387–1398, 2000.

We have also changed "On the other hand" to "Additionally" in the revised MS.

Organization of Section 3.2 and Line 205: I think Line 205 (the latent heat argument) would be more effective in the explanation for why ROS matters to melt onset, which is explored in the prior section. Additionally, Section 3.2 is short enough, that I think the authors would be better off by basically flipping the order of the first two paragraphs. Paragraphs 1 and 3 cover similar ground and could be condensed a little if consecutive. Another reason for this is that the last line of Section 3.3 makes sense more immediately following the current paragraph 1 (179-187) than following paragraph 2 (189-196). Response: Done.

Line 251: "Canadian Basin" is used here, but "Canada Basin" is used elsewhere. Response: We have used 'Canada Basin' uniformly throughout the revised text.

Lines 260-261: I think the authors can use stronger language here – in all eight cases, the trends are predominantly positive, and that's not clear from the current statement. Response: We have modified the statement here based on your suggestion. "Overall, the RPR trends are predominantly positive over the Arctic Ocean in May and June, although there are large spatial differences among the products in both months."

Lines 269-272: This statement goes by fast, and the authors don't fully explain what "further analysis" was done. Looking back at the methods section, this must be the Bintanja (2018) method. A brief reminder here would go a long way. Response: This statement has been removed in the revised MS since it will not affect the main conclusion of this study, as in the reply to your previous comment.

Line 280: I am not sure what the phrasing "got contributions of the largest decreasing trend" means. Do the authors simply mean these regions "exhibit" the largest decreasing trend"? Response: This has been revised to: "The Kara/Barents Seas and Canadian Arctic Archipelago exhibit the largest decreasing trend (more than -2.0cm/decade)."

Line 285: A follow-up of sorts to the Webster et al. (2014) paper also explored the importance of cyclone activity in driving snow depth (Webster et al., 2019). This might be a good place to discuss that. Here or perhaps in the introduction. Response: Thank you for your reminder, we did not notice this reference in previous research. We have added a statement of the long-term changes in the surface snow of sea ice and its causes in the "Introduction" section (L326-327) in the revised MS and also revised the statement here (L339). "Webster et al. (2019) investigated the inter-decadal changes in snow depth over Arctic sea ice, and attributed its variability and trends mainly to cyclone activity and accompanying precipitation, followed by the sea-ice freeze-up." "This study suggests that the interdecadal decrease in snow depth on sea ice in spring is enhanced by the change in precipitation phase (solid to liquid) during the initial ablation period, in addition to the impacts from variability in cyclone snowfall over the snow accumulation season (Webster et al., 2019) and delayed sea ice freeze-up during autumn (Webster et al., 2014)."

Section 4 Comments Lines 291-292: In isolation, this statement is not true because other studies have combined coastal station observations and renalyses in North America to detect rain-on-snow events (e.g., Rennert et al., 2009; Bieniek et al., 2018; Crawford et al., 2020). I believe the spirit of this statement is that this is the first study to synthesize these datasets to examine rain on snow-covered sea ice. I agree that the application to sea ice is novel, but tweaking the language slightly would be best. Response: Thank you for your comment. We have made clarification here to emphasize that the innovation of this work lies in the application to sea ice. "This study, for

the first time, synthesizes station observations at coastal sites in Arctic North America and multiple atmospheric reanalysis datasets, to examine rain on snow events over sea ice."

Lines 295 – 320: These paragraphs are all strictly summary of the results and add little or no new discussion. They are therefore less effective than the first paragraph or last three paragraphs of this section. I encourage the authors to condense them into one paragraph that highlights one or two key points in a more generalized way (i.e., shifting closer to the language of the abstract). Response: Done.

Lines 305-306: In Lines 227-228, the authors mention a "lack of significant trends" for March and April, so it seems incongruous to mention them here as increasing trends. I would restate this as "May and June" instead of "March-June". Response: Thank you for your comment. We have corrected the statement here. "The rain-precipitation-ratio (RPR) averaged over the Arctic Ocean shows a significant increasing trend in May and June in all reanalysis datasets, although there are differences in the magnitude of the trend among the datasets."

It also might make sense to remove "The RPR value and its increasing trend averaged over the Arctic Ocean are strongest in May and June in all datasets," if only May and June are discussed. Response: Done.

Lines 322-346: I think these paragraphs are stronger than the earlier part of this section. The first sentence of the penultimate paragraph in particular is a clear, useful summative sentence. But the length of this discussion/conclusions section buries that statement. Response: We have adjusted the structure of Section 4 according to your suggestions, and deleted and summarized several detailed and specific conclusions. The revised section is more compact, which should be helpful to highlight the main conclusions.

Works Cited in this Review: Bieniek, P. A., Bhatt, U. S., Walsh, J. E., Lader, R., Griffith, B., Roach, J. K., & Thoman, R. L. (2018). Assessment of Alaska rain-on-snow events

using dynamical downscaling. Journal of Applied Meteorology and Climatology, 57(8), 1847–1863. http://doi.org/10.1175/JAMC-D-17-0276.1 Boisvert, L. N., Webster, M. A., Petty, A. A., Markus, T., Bromwich, D. H., & Cullather, R. I. (2018). Intercomparison of Precipitation Estimates over the Arctic Ocean and Its Peripheral Seas from Reanalyses. Journal of Climate, 31(20), 8441–8462. http://doi.org/10.1175/JCLI-D-18-0125.1 Crawford, A. D., Alley, K. E., Cooke, A. M., & Serreze, M. C. (2020). Synoptic Climatology of Rain-on-Snow Events in Alaska. Monthly Weather Review, 148, 1275–1295. http://doi.org/10.1175/MWR-D-19-0311.s1 Kapsch, M.-L., Graversen, R. G., Tjernström, M., & Bintanja, R. (2016). The effect of downwelling longwave and shortwave radiation on Arctic summer sea ice. Journal of Climate, 29(3), 1143–1159. http://doi.org/10.1175/JCLI-D-15-0238.1 Liu, Z., & Schweiger, A. (2017). Synoptic conditions, clouds, and sea ice melt-onset in the Beaufort and Chukchi Seasonal Ice Zone. Journal of Climate, 30, 6999–7016. http://doi.org/10.1175/JCLI-D-16-0887.1 Markus, T., & Cavalieri, D. J. (2000). An enhancement of the NASA Team sea ice algorithm. IEEE Transactions on Geoscience and Remote Sensing, 38(3), 1387–1398. http://doi.org/10.1109/36.843033 Reichle, R. H. Q. Liu, R. D. Koster, C. S. Draper, S. P. P. Mahanama, and G. S. Partyka, (2017). Land surface precipitation in MERRA-2. Journal of Climate, 30, 1643–1664, https://doi.org/10.1175/JCLI-D- 16-0570.1. Rennert, K. J., Roe, G., Putkonen, J., & Bitz, C. M. (2009). Soil thermal and ecological impacts of rain on snow events in the circumpolar Arctic. Journal of Climate, 22(9), 2302–2315. http://doi.org/10.1175/2008JCLI2117.1 Webster, M. A., Parker, C., Boisvert, L., & Kwok, R. (2019). The role of cyclone activity in snow accumulation on Arctic sea ice. Nature Communications, 10(1), 1–12. http://doi.org/10.1038/s41467-019-13299-8.

Please also note the supplement to this comment:
https://tc.copernicus.org/preprints/tc-2020-214/tc-2020-214-AC1-supplement.pdf

---

## Author Comment (AC3) · 9 Dec 2020

Dear Editor, We have studied the valuable comments from yourself and the reviewers carefully, and made further revisions in the manuscript that address your and the reviewers' concerns. Our detailed response to the reviewers' comments follows below. - Response to reviewer#2's comments: This study evaluates the performance of four reanalysis datasets (ERA-I, JRA-55, MERRA2, and ERA5) in representing the timing of ROS events and the phase change of precipitation during the spring melt season over the Arctic Ocean. Comparing with observations at 15 Arctic coastal weather stations, the authors find that the date of the first ROS events in ERA-I is closer to the obser-

vations than that in the other three products, while ERA5 better represents the phase change of precipitation associated with ROS events in spring. The study then investigates trends and spatial variations of ROS events and rain-precipitation-ratio (RPR) over the Arctic Ocean during the melt season. The results show trends towards earlier spring ROS events over most of the Arctic Ocean in recent decades, with the most negative trends in the marginal seas. There has been a clear transition from solid to liquid precipitation over the Arctic Ocean from Match to June, consistent with more ROS events in spring. General Comments This is an interesting study and fits well to The Cryosphere. Overall the paper reads well, however, I have one major concern, which is the weak justification of the use of ERA-I and ERA5 for trend analyses. In addition, the structure of the paper is not always clear. Outlined in the comments below are some suggestions that will hopefully improve the final version of the paper. 1) Why the four reanalysis products (ERA-I, JRA-55, MERRA2, and ERA5) are chosen for the study? Especially MERRA2 instead of MERRA? Boisvert et al. (2018) suggested that precipitation in MERRA was realistic but there were large biases in MERRA2. In their abstract: "When compared with drifting ice mass balance buoys, three reanalyses (ERA-Interim, MERRA, and NCEP R2) produce realistic magnitudes and temporal agreement with observed precipitation events, while two products [MERRA, version 2 (MERRA-2), and CFSR] show large, implausible magnitudes in precipitation events."

Response: Thank you for your comments. This study was conducted based on the global reanalysis datasets those can distinguish between solid and liquid precipitation. According to the results of Boisvert et al. (2018), we select three representative datasets: ERA-I, JRA-55 and MERRA. ERA5 represents the new generation of ECMWF atmospheric reanalyses, which was included in this study since it has not been evaluated in previous studies. Boisvert et al. (2018) have evaluated the advantages and disadvantages of MERRA and MERRA-2, and compared their performance in simulating the precipitation changes in the Arctic Ocean. In order to avoid unnecessary repetitive work, this study chose one of them as the representative of NASA products according to Boisvert et al. (2018) and our own analysis. We compared their

performance in simulating the first rainfall date and RPR. Results show that the average value and spatial distribution of the first rainfall date are basically the same in these two reanalysis datasets, which is consistent with the results of Boisvert et al. (2018), in their work, they suggested that the two datasets have highly consistent descriptions of the rainfall days in the Arctic Ocean. In addition, both of the datasets overestimate snowfall and underestimate rainfall in the Arctic Ocean (Fig. 5, and Fig. 11-12 in Boisvert et al., 2018), leading to an underestimation of RPR in the Arctic Ocean (Fig. R1 and Fig. 4). This situation is more serious in MERRA-2 (Fig. R1). As response to the second comment from the Reviewer#1, we have rechecked the data used in this work, and found that we actually used MERRA data in this study. We mistakenly wrote it as MERRA-2 in the original MS. As show in Figure R1, MERRA-2 gives a lower RPR because it overestimates the snowfall too much and underestimated the rainfall in the Arctic Ocean. We corrected this in the revised MS and added a discussion about the results of Boisvert et al. (2018). Please see details at L83-90 in the revised MS with trace of changes. "Boisvert et al. (2018) evaluated the performance of various reanalysis datasets in simulating the precipitation in the Arctic Ocean, and showed that Modern-Era Retrospective analysis for Research and Applications (MERRA version 2, Gelaro et al. 2017) significantly overestimates the total precipitation compared to MERRA (Rienecker et al., 2011), ERA-I and JRA-55. They further pointed out that MERRA and MERRA-2 have both overestimated snowfall, especially for MERRA-2. In contrast, they have a significant underestimation of rainfall (both for rainfall amount and rainfall days), leading to an underestimation of RPR in the Arctic Ocean. Accordingly, we chose MERRA for the analysis in this study since its underestimation is relatively slight." Figure R1ïïjŽTime series of monthly rainfall precipitation ratio (RPR) averaged over the Arctic Ocean for MERRA and MERRA-2.

2) Observations from 15 coastal stations are used to assess the performance of the reanalysis in this study. In my opinion, the observations at the coastal stations may be representative of nearby marginal seas, but unlikely to be representative of the central Arctic Ocean. Therefore the justification of using ERA-I for ROS events and ERA5

for RPR analyses is rather weak in the paper. The authors should at least provide a summary of evaluation results from previous studies, i.e. Boisvert et al. (2018). In addition, it is critical for the reanalysis products to be consistent over time (1980-2017) for trend analyses. Please verify this in the paper.

Response: Thank you for your suggestion. Current long-term precipitation observation data only exist in the land area of the Arctic, and the limited records obtained by IMB buoy in sea ice area cannot be used to verify the long-term trends of precipitation. Therefore, this study applied the precipitation data from the coastal stations closest to the sea ice area to validate the reanalysis dataset. According to the reviewer's suggestion, we have included a summary of evaluation results from Boisvert et al. (2018) in the revised MS. Please see the response to your first comment. Thank you for your reminder, we have checked the time series used in the full text, including those for trend analysis, and modified Fig. 4d and related content in the revised MS.

Specific comments Section 2.1, Please provide a bit more detail about the reanalysis products, such as the resolutions. Map in Fig. 2(b) is somewhat blurry, is that due to the coarser resolution of JRA-55 relative to the other three products?

Response: Thank you for your suggestion. More detailed description has been included in the revised MS (L92-100): "ERA-I uses the ECMWF forecasting model [version cycle 31r1 (CY31r1)] with a horizontal resolution of T213 ($\sim$78 km). ERA5 is the fifth generation reanalysis from ECMWF. It provides several improvements compared to ERA-I, as detailed by Hersbach and Dee (2016). The analysis is produced at a 1-hourly time step using a significantly more advanced 4Dvar assimilation scheme with a horizontal resolution of approximately 30km. JRA-55 is the first atmospheric global reanalysis dataset and covers a period extending back to 1958. It is based on the TL319 (55km$\times$55km) spectral resolution version, with linear Gaussian grid, of the JMA global spectral model (GSM) with 4DVAR and also incorporates TOVS and SSM/I satellite data. MERRA uses the Goddard Earth Observing System Data Assimilation System (GEOS-5) (Rienecker et al. 2008). It applies the GEOS-5 AGCM

dynamical atmospheric model, which includes a finite-volume dynamical core and a native latitude–longitude horizontal resolution of $1/2°×2/3°$." Regarding the problem in figure2b you mentioned, we found that the "blurry" is caused by the selected format of image when it was generated by NCL. We have corrected this figure in the revised manuscript. Thank you so much for your reminder.

L83-84, are the results sensitive to the 0.5 mm threshold used to determine the occurrence of rainfall? I suggest the authors do some tests on this if haven't already. Ideally, the results shouldn't be too sensitive to the threshold.

Response: Thank you for your suggestion. The reanalysis data generally overestimates the number of trace precipitation (Dai et al., 2006; Boisvert et al., 2018), so the frequency and occurrence time of precipitation event in the reanalysis are sensitive to changes in the precipitation threshold. For this study, different precipitation thresholds may mainly have an impact on the date of the first rainfall event. We have tried different precipitation thresholds, and found that when the threshold exceeds 0.5mm/day, the changes of different precipitation thresholds have negligible effect on the timing of first ROS event and the spatial distribution of its trend in various reanalysis datasets, including MERRA. Our analysis showed that when the threshold is set to 0.5mm/day, the date of first ROS event given by MERRA is consistent with the results of ERA5 and JRA-55, and the spatial pattern of trend of first ROS date is similar with ERA-I and JRA-55. Therefore, we use this value as a threshold in this study. As in the response to review#1, we clarified this in the revised MS (L181-186): "Changing the threshold of precipitation events may affect the determination of the date of the first rainfall to a certain extent, especially for the reanalysis with more frequent trace precipitation, such as MERRA and MERRA-2 (Boisvert et al. al., 2018). Therefore, a lower precipitation threshold may result in an earlier ROS event in these reanalysis datasets, while a higher threshold may result in a later ROS event. However, different thresholds will not have a fundamental impact on the spatial distribution of the trend of the first rainfall timing." Reference: Boisvert, L. N., Webster, M. A., Petty, A. A. et

al.: Intercomparison of precipitation estimates over the Arctic Ocean and its peripheral seas from reanalyses, J. Clim., 31, 8441–8462, 2018. Dai, A., 2006: Precipitation characteristics in eighteen coupled climate models. J. Climate, 19, 4605–4630, https://doi.org/10.1175/JCLI3884.1.

L97-98, "The sea ice extent is calculated from SIC using a threshold of 15%", please provide a reference.

Response: Thank you for your suggestion. We have included an earlier study that used the threshold of 15% in the revised MS. "Gloersen, P., Campbell, W. J., Cavalieri, D. J., Comiso, J. C., Parkinson, C., and Zwally, H. J.: Satellite passive microwave observations and analysis of Arctic and Antarctic sea ice, 1978-1987, Annals of Glaciology, 17, 149-154, 1993."

Section 2.3, Please explain why observations at other weather stations are not included, such as those along the coast of European Arctic.

Response: Thank you for your comment. The precipitation observations along the coast of European Arctic available to us are mainly located in the Nordic arctic region-northern Norway, Sweden, and Finland, including the arctic islands Svalbard and Jan Mayen (Førland et al., 2020; Vikhamar-Schuler et al., 2016). The stations in Svalbard region are located on the west coast of Spitsbergen or on the south to Spitsbergen. During our study period, there was no sea ice near the stations in the Nordic arctic region. Thus, they are not included in this study. However, we added a discussion on the results of previous studies in Svalbard in the revised MS (L307-313). "There is also an increasing trend close to sea ice edges in the Atlantic sector, especially in the Nordic arctic region. A recent study based on the station observations in Svalbard demonstrates that the solid precipitation has decreased at a rate of 2.3-6.5 % per decade in this region during the past decades, while the liquid precipitation has increased at a rate of 0.6-9.4 % per decade during the same period (Førland et al. 2020). This is generally consistent with our results in the Atlantic sector (Fig. 4-5)." Reference: Førland,

E. J., K. Isaksen, J. Lutz, I. Hanssen-Bauer, T. V. Schuler, A. Dobler, H. M. Gjelten, and D. Vikhamar-Schuler, 2020: Measured and Modeled Historical Precipitation Trends for Svalbard. J. Hydrometeor., 21, 1279–1296, https://doi.org/10.1175/JHM-D-19-0252.1. Peeters B, Pedersen ÅØ, Loe LE, Isaksen K, Veiberg V, Stien A, Kohler J, Gallet J-C, Aanes R, Hansen BB. 2019. Spatiotemporal patterns of rain-on-snow and basal ice in high Arctic Svalbard: detection of a climate-cryosphere regime shift. Environmental Research Letters. 14 015002, https://doi.org/10.1088/1748-9326/aaefb3. Vikhamar-Schuler D, Isaksen K, Haugen JE, Tømmervik H, Luks B, Schuler T, Bjerke J. 2016. Changes in winter warming events in the Nordic Arctic Region. Journal of Climate 29, doi: 10.1175/JCLI-D-15-0763.1.

Fig.2, it would be helpful to provide some comments about the positive/later trends occurring in (a)-(c), but not in (d).

Response: During the transition period between spring and summer, the earlier ROS events in most parts of the Arctic Ocean (Fig. 2) could be attributed to the rapid warming of the Arctic. During this period, the climate began to warm up, and the local temperature rose to around melting point by May and June. Therefore, the precipitation phase during this period is very sensitive to climate warming. However, we cannot explain the reason for the unusual delay of the first ROS event in Atlantic and Pacific sectors in ERA-I, MERRA and JRA-55 because the air temperature rise is significant in these areas. In-depth study is needed in the next step.

Section 3.2, I'd suggest to move the first paragraph to the Introduction. I think this would make the paper tighter and make it easier for readers to have a better understanding of the linkages between different components of the paper.

Response: Thank you for your suggestion. However, is it better to keep this paragraph here because Section 3.2 is short enough? We have flipped the order of the first two paragraphs since Paragraphs 1 and 3 cover similar ground and could be condensed a little if consecutive. Please see details at L214-232, or see our response to the

comment for section 3.2 by Reviewer#1.

Fig.3, I wonder if the significant correlations between EMO and ROS are due to the fact that they are both correlated with surface air temperature (SAT)? SAT was used to indicate melt onset on sea ice in previous studies. Dou et al. (2019) suggested that the year-to-year variability of the timing of first spring ROS was closely tied to the timing of persistent warming events.

Response: Thank you for your comment. In the early period, people generally believed that the temperature rise was the main factor resulting in the sea ice melt onset. With the enrichment of observational data, people found that there is no good correspondence between sensible heat (warm air) and melt onset, although their long-term trends are consistent (e.g., Persson, 2012; Mortin et al., 2016 ; Dou et al., 2019). Moreover, liquid precipitation did not always occur during every warm event (Vikhamar-Schuler et al., 2016). Therefore, there is not a good correspondence between SAT and ROS events too. In recent years, the melt onset was found to be triggered by moist, warm air masses associated with synoptic-scale weather systems that augmented the atmospheric energy fluxes to the surface (Persson, 2012). Mortin et al. (2016) further showed that melt onset over Arctic sea ice is initiated by positive anomalies of water vapor, clouds, and air temperatures that increase the downwelling longwave radiation (LWD) to the surface. The earlier melt onset occurs; the stronger are these anomalies. When melt occurs early, an anomalously opaque atmosphere with positive LWD anomalies preconditions the surface for weeks preceding melt. In contrast, when melt begins late, clearer than usual conditions are evident prior to melt. However, the previous studies have not evaluated the impact of rainfall (ROS event) on the melt onset. Note that ROS events may occur alongside abundant water vapor and clouds (e.g., Bieniek et al., 2018). In other words, while the LWD generated by clouds and water vapor continues to heat the ground, it may also be accompanied by ROS events that could initiate the sea ice ablation (Dou et al., 2019). Our study estimated the sensitivity of sea ice melt onset to on large spatial scale or the possible connection between
them. Further research on the exact role of liquid precipitation in triggering sea ice melt onset is needed based on large-scale observations of precipitation and melt onset. Reference: Dou, T., Xiao, C., Liu, J. et al.: A key factor initiating surface ablation of Arctic sea ice: earlier and increasing liquid precipitation, The Cryosphere, 13, 1233–1246, https://doi.org/10.5194/tc-13-1233-2019, 2019. Mortin, J., G. Svensson, R. G. Graversen, M.-L. Kapsch, J. C. Stroeve, and L. N. Boisvert: Melt onset over Arctic sea ice controlled by atmospheric moisture transport, Geophys. Res. Lett., 43, 6636–6642, doi:10.1002/ 2016GL069330, 2016. Persson, P. O. G. : Onset and end of the summer melt season over sea ice: Thermal structure and surface energy perspective from SHEBA, Clim. Dyn., 39(6), 1349–1371, doi:10.1007/s00382-011-1196-9, 2012. Vikhamar-Schuler, D., Isaksen, K., Haugen, J.E., Tømmervik, H., Luks, B., Schuler, T., Bjerke, J.: Changes in winter warming events in the Nordic Arctic Region. Journal of Climate 29, doi: 10.1175/JCLI-D-15-0763.1, 2016.

L195, FRD is not defined.

Response: Thank you for your reminder. The definition of 'FRD' has been included in the section of 'Methods' in the revised MS (L152).

Section 3.3, I'd suggest move the first paragraph to the Introduction, see above.

Response: We merged part of the content in the first paragraph into the second paragraph and deleted the rest. Please see details at L250-259. "The amount of rainfall depends on the total precipitation and the portion of total precipitation occurring as rainfall, as quantified by the rain precipitation ratio, RPR (see section 2.5). Below, we evaluated and compared changes in RPR across four different reanalysis products. The RPR averaged over the Arctic Ocean is overall higher for all spring months in ERA-I than in the other three reanalysis datasets (Fig. 4)."

L317, ESAO is not defined.

Response: This has been defined at L295 in the revised MS.

L321-330, "ERA5 more reasonably reproduces the observed RPR and its trends than ERA-I compared with station observations. Several new techniques have been incorporated into ERA5: : :", this seems to be in contrast with the large negative bias for first ROS events in ERA5 relative to ERA-I and observations shown earlier (L150), can you explain why?

Response: Thank you for your comment. The first rainfall date can be determined even if a light rain occurs. Therefore, an earlier ROS event does not mean a greater rainfall during the early melt season, because the amount of rainfall in a time window often depends on several large precipitation events, which is not directly related to the timing of first rainfall.

Fig.5, I'd prefer to have maps in May on the upper panel and June on the bottom panel. Please include the name of the reanalysis product used for the results in Fig.3 and 7 in the captions.

Response: Done.